# Micromixing Intensification within a Combination of T-Type Micromixer and Micropacked Bed

**DOI:** 10.3390/mi14010045

**Published:** 2022-12-24

**Authors:** Zhou Lan, Yangcheng Lu

**Affiliations:** State Key Laboratory of Chemical Engineering, Department of Chemical Engineering, Tsinghua University, Beijing 100084, China

**Keywords:** micromixing, micropacked bed, microstructual combination, viscous system

## Abstract

The combination of microstructural units is an effective strategy to improve the micromixing of liquid phase systems, especially viscous systems. However, how the microstructural combination influences micromixing is still not systematically investigated. In this work, the Villermaux/Dushman reaction is used to study the micromixing performance of the viscous system of the glycerol–water in the combination of a T-type micromixer and a micropacked bed. Micromixing performances under various structural parameters and fluid characteristics are determined and summarized, and the micromixing laws are revealed by dimensionless analysis considering the specific spatial characteristics and temporal sequence in the combined microstructures. It achieves good agreement with experimental results and enables guidance for the design and scaling-up of the combined T-type micromixer and micropacked bed towards micromixing intensification in viscous reaction systems.

## 1. Introduction

The liquid phase system is widely involved in chemical processes [1,2,3]. Typical liquid–liquid reaction systems, such as nitration [4,5,6,7], saponification [8,9], and polymerization [10,11,12,13], were commonly sensitive to mass transfer involving micromixing, depending on the viscosity and the ratio of flow rates remarkably [1,14]. In conventional stirring equipment, low mass transfer efficiency, as well as micromixing performance, were always a limit of these liquid phase chemical processes.

The continuous flow microreactor technologies developed in recent years are powerful tools to overcome the problem of poor micromixing due to their high specific surface area and also present excellent mass and heat transfer performance and intrinsic safety [1,3,15,16,17,18]. Many strategies to enhance the micromixing performance have also been derived for microreactors, most of which belong to passive mixing without external energy input through intensifying diffusion or chaotic advection, such as special channel structures [19,20,21,22], microbubble enhanced mixing [14,23], and micropacked bed reactors [24,25,26]. At the same time, many methods to characterize the micromixing performance have also been developed, including the visualization methods and the chemical molecular probe methods [27,28]. The visualization methods mainly use the dye tracer or the acid–base indicator to characterize the micromixing performance [29,30], which has the advantage of perceptual intuition but is limited by image capture and analysis capacity. Comparatively, the chemical probe methods are easy to implement and quantify, which introduce probe reactions such as the Villermaux/Dushman reaction [25,31,32,33], the fourth Bourne reaction [34,35,36], acid–base reaction [37,38,39,40], and redox reaction [41,42,43]. Additionally, the numerical simulation may give more clues about flow physics for micromixing performance [44,45]. However, as for mixing flow in complicated spatial structures and extended spatio-temporal scale, a reliable numerical methodology was still challenged in terms of geometric modeling and computational force and is worth further developing.

The liquid–liquid micromixing is challenged by the incremental differences between the mixing fluids in terms of viscosity, density, and wettability [1,14,46]. In particular, the low diffusion coefficient in high-viscosity fluids resulted in slow molecular motion [14], and increasing the viscosity also decreased the Reynold number and inhibited the generation of convection individually. Therefore, it is hard to achieve fast and complete mixing and to provide continuous mixing enhancement with conventional simple micromixing elements such as T-type micromixers. As an alternative, a microstructural combination was proposed to enhance the mixing of viscous fluids. In the previous research by our group [4,47], the combination of a T-type micromixer and a micropacked bed reactor was successfully used to enhance the liquid phase reaction in viscous systems. However, the quantitative studies on the micromixing characteristics of this combination, as the fundamental of process and equipment design, were still limited.

In this work, we aim to quantitatively evaluate and interpret the micromixing performance of viscous systems in the combination of a T-type micromixer, a connecting pipe, and a micropacked bed. By using all these elements that can be obtained commercially, we tried to establish a quantitative methodology on the design of combinatory, which is various and necessary for enhancing the micromixing of viscous systems. For this purpose, the Villermaux/Dushman reaction was adopted as a characterization method, and the glycerol–water solutions as the model viscous systems. The influences of microstructure parameters, fluid viscosities, and the ratio of flow rates of mixing fluids were systematically investigated. By considering the connection of various microstructure elements for micromixing based on their spatio-temporal relationship, the dimensionless correlation was also established for micromixing performance prediction and combinatory design.

## 2. Materials and Methods

Analytical grade potassium iodide (KI, ≥99%; Aladdin, Shanghai, China), potassium iodate (KIO_3_, ≥99.8%; Aladdin, China), boric acid (H_3_BO_3_, ≥99.5%; Aladdin, China), iodine (I2, ≥99.8%; Aladdin, China), sodium hydroxide (NaOH, ≥96%; Greagent, China), and glycerol (GL, ≥99%; Greagent, China) were used directly without further purification. Water (resistivity of 18.2 MΩ·cm) used throughout the experiment was prepared by an ultrapure water system (Center 120FV-S, The lab, Haverhill, MA, USA).

The Villermaux/Dushman reaction system was used to characterize the micromixing performance [27], which included a group of parallel competitive reactions as follows [31,32]:(1)$$H_{2}{\mathrm{BO}}_{3}^{-}{+H}^{+}\;\overset{k_{1}}{\to }{\;H}_{3}{\mathrm{BO}}_{3}$$
(2)$${5I}^{-}{+\mathrm{IO}}_{3}^{-}{+6H}^{+}\;\overset{k_{2}}{\to }{\;3I}_{2}{+3H}_{2}O$$
(3)$$I_{2}{+I}^{-}\;\overset{K_{C}}{↔}{\;I}_{3}^{-}$$

Among them, reaction (1) can be considered an instantaneous reaction, reaction (2) is a fast reaction, and reaction (3) is an equilibrium reaction with an equilibrium constant expressed as:(4)$$K_{C}=\frac{C_{I_{3}^{-}}}{C_{I_{2}}{\cdot C}_{I^{-}}}$$ K_C_ can be obtained from the empirical correlation with temperature [48,49]:(5)$${\mathrm{lgK}}_{C}=\frac{555}{T}+7.355\;-\;2.575\cdot \mathrm{lgT}$$

When the micromixing is ideal, the acid is distributed evenly instantly, and H^+^ can be completely consumed by $H_{2}{\mathrm{BO}}_{3}^{-}$ in reaction (1) to form H_3_BO_3_. Otherwise, some H^+^ may participate in a reaction (2) to generate $I_{2}$, and then transform to $I_{3}^{-}$ through reaction (3). It implies that lower $I_{3}^{-}$ concentration corresponds to faster mixing.

The segregation index (*X*_S_), derived from the concentration of $I_{3}^{-}$, was defined to quantitatively characterize the micromixing performance of the system [31,32]. Specifically, the mole ratio of the H^+^ consumed by reaction (2) to the total H^+^, denoted as *Y*, is:(6)$$Y=\frac{{2(n}_{I_{2}}{+n}_{I_{3}^{-}})}{n_{H^{+},0}}$$

If micromixing did not take place at all, reactions (1) and (2) would conduct separately, corresponding to a value of *Y* at the segregation state:(7)$$Y_{\mathrm{ST}}=\frac{{6C}_{{\mathrm{IO}}_{3}^{-},0}}{{6C}_{{\mathrm{IO}}_{3}^{-},0}{+C}_{H_{2}{\mathrm{BO}}_{3}^{-},0}}$$

Additionally, *X*_S_ is the ratio of the actual *Y* to *Y*_ST_:(8)$$X_{S}=\frac{Y}{Y_{\mathrm{ST}}}=\frac{{(Q}_{A}{+Q}_{B}{)(C}_{I_{2}}{+C}_{I_{3}^{-}})}{Q_{B}C_{H^{+},0}}\frac{{6C}_{{\mathrm{IO}}_{3}^{-},0}{+C}_{H_{2}{\mathrm{BO}}_{3}^{-},0}}{{3C}_{{\mathrm{IO}}_{3}^{-},0}}$$
where Q_A_ and Q_B_ are the flow rates of solution A (KI, KIO_3_, $H_{2}{\mathrm{BO}}_{3}^{-}$/H_3_BO_3_ solution) and solution B (acid solution), respectively, and $C_{I_{2}}$ and $C_{I_{3}^{-}}$ can be calculated by the combination of the reaction equations and the mass conservation of iodine, written as Equation (9).
(9)$$-\frac{5}{3}C_{I_{2}}^{2}+\left(C_{I^{-},0}\;-\;\frac{8}{3}C_{I_{3}^{-}}\right)C_{I_{2}}\;-\;\frac{C_{I_{3}^{-}}}{K_{C}}=0$$

Thus, through monitoring $I_{3}^{-}$ concentration in real-time by online UV absorption spectrum, *X*_S_ can be calculated. The variation range of the *X*_S_ is 0~1. *X*_S_ = 1 corresponds to a segregation state without mixing at all; *X*_S_ = 0 corresponds to a completely ideal state of micromixing. *X*_S_ is a typical and easily accessible index for micromixing characterization.

The experimental setup of the micromixing performance characterization is shown in Figure 1. The glycerol–water solutions with different viscosity were prepared according to the relationship between the content of glycerol and the viscosity presented in the physical properties manual [50]. Then, they were bubbled with nitrogen (N_2_) for more than one hour to remove the dissolved oxygen before adding reactants. The concentrations of the reactants are listed in Table 1.

In the experiment, two constant flux pumps (Beijing Xingda Science and Technology Development Co., Ltd., Beijing, China; flow rate, 0~30.00 mL/min; pressure range, 0~20 MPa, repeat precision ≤ ±1%) were used to deliver solution A and solution B to a microstructure combinatory consisting of a T-type micromixer, a connecting pipe, and a micropacked bed. In the T-type micromixer, solution A and solution B were contacted in crossflow mode, and the latter flowed perpendicularly. All experiments were carried out at room temperature (25 °C). The effluent flowed into the online ultraviolet flow cell (FIA-ZSMA-ML-100-PEEK, Ocean Optics), and the 353 nm band was measured by an online ultraviolet device (DH-2000-FHS-DUV-TTL light source, QEPRO fiber spectrometer, Ocean Optics, precision ≤ ±0.4%), which was transformed into the concentration of $I_{3}^{-}$ for *X*_S_ calculation by using the standard curve in Appendix A. In experiments, all the pumps and monitors were calibrated in advance to ensure to work well. Each measurement was repeated five times, and the error bars were recorded.

All valves and tubes (inside diameter (I.D.), 0.75 mm; outer diameter (O.D.), 1.60 mm) were made of 316L stainless steel. Among them, the micromixer was a T-type mixing tee (I.D., 0.25 mm); the micropacked bed reactor was a pipeline (I.D., 4 mm; O.D., 6 mm) filled with glass beads supported by PTFE meshes (0.150 mm) at both ends. Four kinds of glass beads (ASONE Co.) were used, of which the size was 0.105~0.125 mm, 0.177~0.250 mm, 0.350~0.500 mm, and 0.500~0.710 mm, respectively, and the void fraction was determined to be 33%, 34%, 38%, and 40%, respectively.

## 3. Results and Discussion

### 3.1. Micromixing Performance of Various Combined Microstructures

Firstly, we established two microstructural combinations, a T-type micromixer connected with a thin tube and a T-type mixer connected with a micropacked bed, to compare their micromixing performance, and a sole T-type micromixer (including connecting pipe) was used as a reference. In this group of experiments, the viscosity of solution A was fixed at 25 mPa·s, the viscosity of solution B was fixed at 1 mPa·s, and the ratio of flow rates of solution A to solution B was 5:1. As shown in Figure 2a, it can be found that for all three cases *X*_S_ decreased with the increase in the flow rates, corresponding to micromixing performance enhancement due to the enhancement of convection. Compared with the T-type micromixers only, both thin tube and micropacked beds following the T-type micromixers could improve the micromixing performance significantly, and the micropacked bed achieved the best micromixing performance at the same flow rates. For example, when the total flow rate was 15 mL/min, *X*_S_ was 0.00111 for the T-type micromixer connected with a micropacked bed and 0.00254 for the T-type micromixer connected with a thin tube, respectively. Following the primary mixing of the T-type micromixer, the secondary crushing of fluids took place in the micropacked bed, resulting in further collision and mixing, which enhanced the micromixing performance.

Then, we filled the packing with different sizes in the micropacked bed and investigated the effect of packing size on the micromixing performance. As shown in Figure 2b, the smaller the packing size is, the better the micromixing performance is. It indicates that the segmentation effect of packing on the fluids plays a major role in mixing intensification. The packing with a smaller size could provide smaller channels for flow segmentation and enable the micropacked bed to achieve mixing intensification at lower flow rates. For instance, when the total flow rate was 6 mL/min, *X*_S_ with fine packing (0.105~0.125 mm) could reach 0.00424, which was comparable to that of a sole T-type micromixer at 15 mL/min. According to the fluids’ viscosity and the flow capacity, the Reynold number in our experiments was always far less than 2000. Judged from the common criterion, the flow pattern was kept at laminar flow.

Herein, we used the same micropacked bed following different T-type micromixers and connecting tubes to investigate the effects of upstream elements. In detail, we changed the I.D. of the micromixer, the I.D. of the connecting pipe, and the length of the connecting pipe separately. Figure 3a presents the effect of the I.D. of the micromixer. When the total flow rate was 9 mL/min, *X*_S_ corresponding to the micromixers with the I.D. of 0.75 mm, 0.50 mm, and 0.25 mm were 0.00617, 0.00467, and 0.00324, respectively. In general, the smaller the I.D. of the micromixer was, the smaller *X*_S_ was, reflecting the better micromixing performance since the decrease in internal diameter decreases the mass transfer distance and accelerates the mixing. Similarly, as shown in Figure 3b, better mixing performance can be achieved by using a connecting pipe with a smaller I.D. The influence of the length of the connecting pipe was shown in Figure 3c: when the flow rate was relatively high, the *X*_S_ of different connecting pipe lengths were similar; when the flow rate was relatively low, the shorter connecting pipe seemed to correspond to smaller *X*_S_. In fact, mixing took place in every element, including the T-type micromixer, the connecting pipe, and the packed bed. In case we observed a lower total byproduct yield (lower *X*_S_) by adding a further element downstream, it indicates that mixing was not complete in the upper elements. In this specific case, mixing was not completed in the T-type micromixer and the connecting pipe and then finalized in the packed bed. In the case of long connecting pipes, triiodide was predominantly formed in the connecting pipes. Hardly any triiodide would be formed in the more effective mixing unit ‘packed bed’. On the contrary, the shorter the connecting pipes, the lower the triiodide concentration because the byproduct would not form in time in the connecting pipes without enhanced mixing.

Next, we investigated the influence of system viscosity on the mixing performance of the combination of a T-type micromixer and a micropacked bed. Solution A was the same as solution B in viscosity, and the results are shown in Figure 4a. As the viscosity increased from 1 mPa·s to 10 mPa·s, *X*_S_ increased gradually. It was difficult to achieve a good mixing performance even at a high flow rate for the system with a viscosity of 10 mPa·s. Subsequently, the micromixing performance characterizations were conducted with a fixed viscosity of one fluid at 5 mPa·s and changing viscosity of the other fluid. Figure 4b corresponds to fixing the viscosity of solution B, and Figure 4c fixes the viscosity of solution A. It can be found that the viscosity of solution A showed a greater influence on *X*_S_ than the viscosity of solution B. This was because solution A had a larger flow rate and greater influence on the properties of the mixed fluid. More examples of the influence of viscosity on micromixing performance can be found in Appendix A in Appendix A.

The influence of the different ratios of flow rates (*Q*_A_:*Q*_B_) on the mixing performance was also investigated, as shown in Figure 5. It can be found that increasing the ratio of flow rates from 1:1 caused poor mixing performance. For example, setting the total flow rate at 9 mL/min, when the ratio of flow rates increased from 5:1 to 10:1, *X*_S_ increased from 0.00460 to 0.00983. The influence of the ratio of flow rates on the mixing performance may be caused by the distribution of the two fluids in the microreactor [14]: under laminar flow regimes; when the ratio of flow rates was large, the flow rate was low for the fluid from the perpendicular inlet of T-type micromixer, and which may be compressed into a thin layer in the downstream pipe and left larger distance for mass transfer in a continuous fluid. In other words, a long distance was required for the dispersed fluid to diffuse throughout the downstream pipe, corresponding to high *X*_S_. Since the secondary crushing in the micropacked bed can provide the re-collision and mixing of the fluid, the compression of the dispersed fluid may be avoided to some extent, which was expected to improve the mixing performance at a large ratio of flow rates effectively.

### 3.2. Correlation and Prediction of Micromixing Performance

The micromixing performance determined above reflected the synergistic effects of the fluid characteristics and the spatio-temporal structure of the combined microstructures. Herein, we tried to establish a predictive dimensionless correlation with a wide range of applications. Considering the existing microreactor mixing performance prediction correlations in the literature [14,24,46,51] and the above-mentioned micromixing experimental results, we proposed the following equation for correlation.
(10)$$X_{S}=x_{1}{\left(\frac{Re_{A}}{Re_{B}}\right)}^{x_{2}}Re_{c}{}^{x_{3}}Re_{\mathrm{bed}}{}^{x_{4}/\left(x_{5}+t_{c}\right)}$$
where the effects of fluid properties, micropacked bed characteristics, and each microstructure on the micromixing process were reflected by the ratio of the initial Reynolds number of the two fluids ($Re_{A}/Re_{B}$), the Reynolds number in the connecting pipe after the micromixer ($Re_{c}$), the Reynolds number in the micropacked bed ($Re_{\mathrm{bed}}$), and the time for the fluid to enter the micropacked bed from the T-type micromixer ($t_{c}$). Additionally, $x_{1}$~$x_{5}$ are associated parameters; $Re_{c}$ and $Re_{\mathrm{bed}}$ are approximately calculated according to the physical properties of the two fluids after mixing; the calculation equation of $Re_{\mathrm{bed}}$ was shown as follows.
(11)$$Re_{bed}=\frac{\rho_{M}u_{bed}d_{eb}}{\mu_{M}}=\frac{\rho_{M}\left(\frac{u_{0}}{\varepsilon }\right)\left(\frac{4\varepsilon }{\left(1-\varepsilon \right)a}\right)}{\mu_{M}}$$
where $\rho_{M}$ and $\mu_{M}$ are the density and viscosity of the mixed fluid, respectively; $u_{0}$ is the flow rate of the fluid in the empty packing tube; ε is the void ratio of the micropacked bed; and a is the specific surface area of the packing particles.

The least-squares function (lsqcurvefit) in the Matlab software was used to fit the experimental data. The fitting results are shown in Figure 6. The values of parameters were 0.0445 for *x*_1_, 0.174 for *x*_2_, −0.479 for *x*_3_, 0.0156 for *x*_4_, −0.572 for *x*_5_, respectively. The experimental values of *X*_S,exp_ agreed with the predicted values of *X*_S,pre_ well, with the fitting errors (Resnorm value) only 1.33 × 10^−4^. All data fall within ±25%, and the *R*^2^ of the linear fit of *X*_S,exp_ and *X*_S,pre_ was 0.853. The fitting of the correlation to some key influencing factors can be found in Appendix A in Appendix A. Correspondingly, Table 2 shows the ranges of main parameters for experimental data acquisition, including the viscosities and Reynolds numbers of two mixing fluids, the ratio of flow rates of fluid A and fluid B, etc. It should be noticed that this correlation applied to a mixed system with a viscosity ≥2.5 mPa·s, and the flow rate of fluid A was greater than the flow rate of fluid B. At the same time, the type, size, and operating mode of the micromixer would also affect the exponential factor of the fitting. The T-type micromixer used here was an equal-diameter tee (I.D., 0.25 mm), the mixing mode was crossflow mixing, and the flow directions of fluid A and fluid B were horizon direction and vertical direction, respectively.

The obtained correlation parameters reflected the effects of various factors quantitatively. In detail, $x_{2}$ is a positive number, indicating that the increase in the ratio of flow rates and the viscosity ratio of the two fluids will weaken the micromixing. $x_{3}$ is negative, indicating that increasing the Reynolds number in the connecting pipe enhances the micromixing. $x_{4}$ is a positive number, $x_{5}$ is a negative number, and $\left|x_{5}\right|>t_{c}$, so $x_{4}/\left(x_{5}+t_{c}\right)<0$, indicating that increasing the Reynolds number in the micropacked bed can enhance the micromixing. $t_{c}$ reflects the time scale of the mixing state evolution before entering the micropacked bed reaction. Too-large $t_{c}$ will lead that the micropacked bed does not have an influence on the Villermaux/Dushman reaction process. Thus, smaller $t_{c}$ is favorable for the micropacked bed to take action and enhance the micromixing.

## 4. Conclusions

In this work, the micromixing performance of a combination of a T-type micromixer, connecting pipe, and micropacked bed under the viscous systems was determined by the Villermaux/Dushman reaction using the glycerol–water system. Through the exploration of the influence laws of the structural parameters, the fluid viscosities, and the ratio of flow rates, we recognized the synergistic effects of the fluid characteristics and the spatio-temporal structure. Introducing the micropacked bed in a suitable way and time could intensify the micromixing of the viscous system effectively, and the segregation index *X*_S_ was less than 0.005, even at a large ratio of flow rates. Additionally, the correlation equations between the segregation factor and the Reynolds numbers of different stages were established with a good agreement and certain physical significance. Understanding the micromixing characteristics of the combined microstructures composed of a T-type micromixer, connecting pipe, and micropacked bed is beneficial to guide the design and application of structured microreactors in practice.

## Figures and Tables

**Figure 1 micromachines-14-00045-f001:**
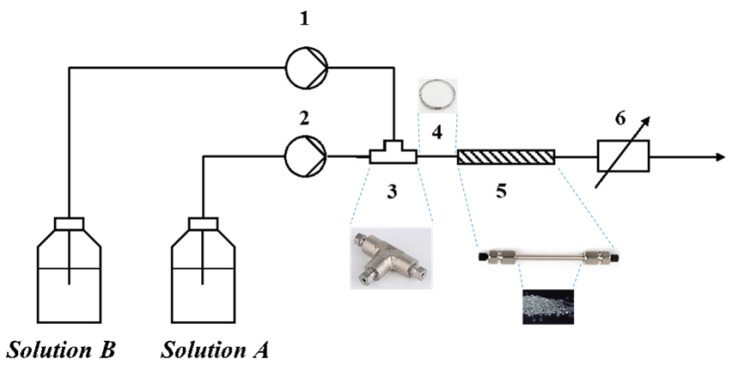
Schematic of the micromixing performance characterization setup: 1 and 2: Delivery pumps; 3: T-type micromixer; 4: Connecting pipe; 5: Micropacked bed reactor; 6: Online UV spectrometer.

**Figure 2 micromachines-14-00045-f002:**
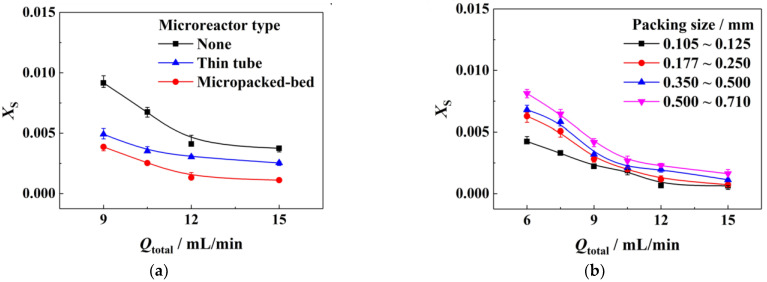
Flow rate profiles of the micromixing performance under various conditions: (**a**) different reactor types; (**b**) different packing sizes. Reference conditions: micromixer, 0.25 mm of I.D.; connecting pipe, 0.5 mm of I.D., 5 cm of length; microreactor, 0.5 mL of empty volume; thin tube, 0.75 mm of I.D., 0.5 mL of empty volume; *Q*_A_:*Q*_B_ = 5:1; *μ*_B_ = 1 mPa·s.; (**a**) packing size, 0.177~0.250 mm, and *μ*_A_ = 25 mPa·s; (**b**) *μ*_A_ = 10 mPa·s.

**Figure 3 micromachines-14-00045-f003:**
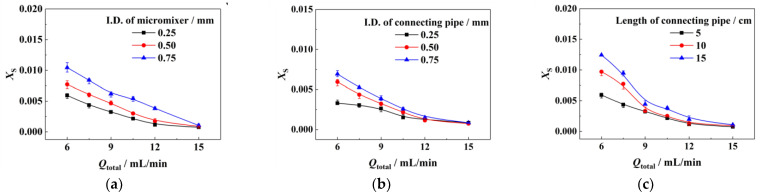
Influence of upstream microstructural elements on micromixing performance: (**a**) I.D. of micromixer, (**b**) I.D. of connecting pipe, and (**c**) length of connecting pipe. Reference conditions: micromixer, 0.25 mm of I.D.; connecting pipe, 0.5 mm of I.D., 5 cm of length; micropacked bed, 10 cm of length; packing size, 0.177~0.250 mm; *Q*_A_:*Q*_B_ = 5:1; *μ*_A_ = 10 mPa·s and *μ*_B_ = 1 mPa·s.

**Figure 4 micromachines-14-00045-f004:**
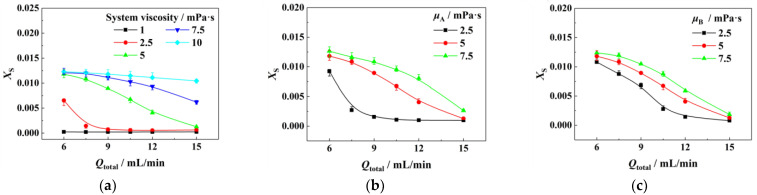
Influence of fluids characteristics on micromixing performance: (**a**) bulk viscosity, (**b**) continuous fluid viscosity, and (**c**) dispersed fluid viscosity. Reference conditions: micromixer, 0.25 mm of I.D.; connecting pipe, 0.5 mm of I.D., 5 cm of length; micropacked bed, 5 cm of length; packing size, 0.177~0.250 mm; *Q*_A_:*Q*_B_ = 5:1. (**a**) *μ*_A_ = *μ*_B_, (**b**) *μ*_B_ = 5 mPa·s, (**c**) *μ*_A_ = 5 mPa·s.

**Figure 5 micromachines-14-00045-f005:**
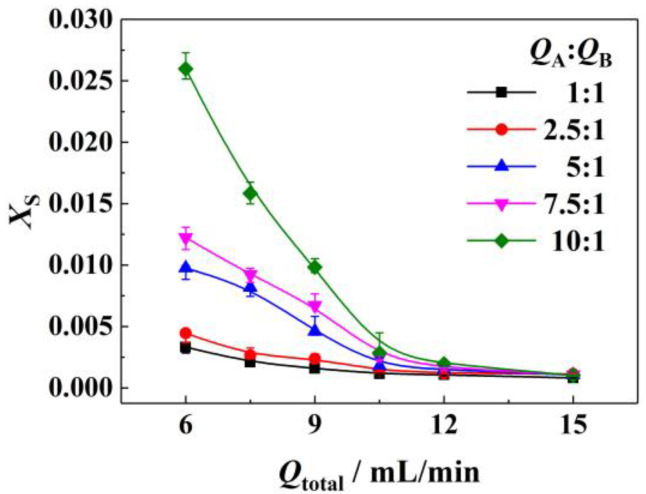
Influence of the ratio of flow rates on micromixing performance. Reference conditions: micromixer, 0.25 mm of I.D.; connecting pipe, 0.5 mm of I.D., 5 cm of length; micropacked bed, 5 cm of length; packing size, 0.177~0.250 mm; *μ*_A_ = 10 mPa·s, *μ*_B_ = 1 mPa·s.

**Figure 6 micromachines-14-00045-f006:**
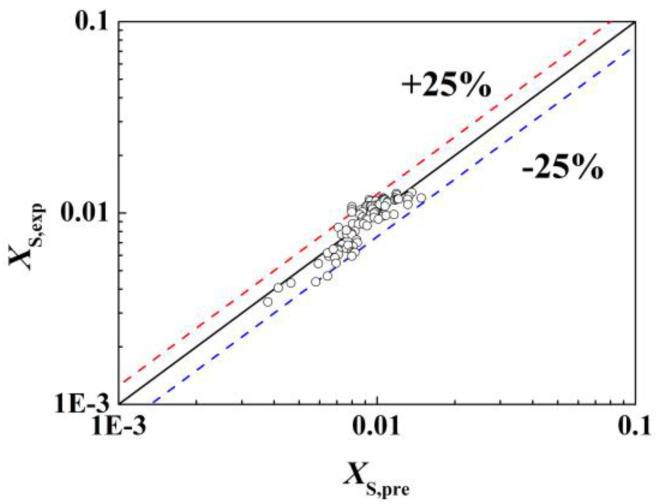
Fitting of liquid–liquid micromixing performance in viscous systems. Black line is the diagonal line that *X_S_*_,exp_ equals to *X_S_*_,pre_. Red line and blue line correspond to +25% and −25% deviation of *X_S_*_,exp_ from *X_S_*_,pre_.

**Table 1 micromachines-14-00045-t001:** Concentration of each reactant in the Villermaux/Dushman reaction system.

	Reactant	Concentration/mol/L
Solution A	KI	0.03
	KIO_3_	0.006
	H_3_BO_3_	0.09
	NaOH	0.09
Solution B ^a^	H_2_SO_4_	0.005

^a^ The concentration of H_2_SO_4_ in solution B was the concentration when the ratio of flow rates of two fluids was 1:1. When the ratio of flow rates of mixing fluids was R:1 (solution A over solution B), the concentration of H_2_SO_4_ in solution B should be (0.005 × R) mol/L.

**Table 2 micromachines-14-00045-t002:** Fitting range of liquid–liquid micromixing correlation in viscous system.

Items	Data
*μ*_A_/mPa·s	2.5~25
*μ*_B_/mPa·s	1~10
Re_A_	6.73~100
Re_B_	1.35~84.6
Q_A_:Q_B_	1:1~10:1
d_p_/mm	0.105~0.710
t_c_/s	0.016~0.530

## Data Availability

The data that support the findings of this study are available within the article and its Appendix A.

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
