# Peer review of "Micromixing Intensification within a Combination of T-Type Micromixer and Micropacked Bed"

_micromachines, 2022, doi:10.3390/mi14010045_

Round 1
Reviewer 1 Report
In this study, the authors focused on micromixing within a combination of T-type micromixer and micropacked bed. They presented experimental results. Major comments are:
1-The micromixer is not displayed in a detailed manner. Drawings and improtant dimensions are missing. Also, novel features of this mixer are not included.
2-Validation efforts are missing in this study. The authors should validate their setup and methodology. Also, uncertainties in the major parameters should be included.
3-The authors should also include numerical results. In this format, flow physics is missing so that enhancement mechanisms could not be explained. Numerical results will give more clues for flow physics and performance enhancements.
4-The authors should include comparisons with the state of art micromixers. Some of passive ones including inertial ones are present in the literature. The authors could benefit from the technical content in such articles.
5- The trends in the experimental results (figs. 2-5) depend on the micromixer and are continuously changing. The changes in trends should be well explained.
Author Response
Responses to Reviewer’s comments
In this study, the authors focused on micromixing within a combination of T-type micromixer and micropacked bed. They presented experimental results. Major comments are:
Comment 1: The micromixer is not displayed in a detailed manner. Drawings and important dimensions are missing. Also, novel features of this mixer are not included.
Reply: Thank you for your suggestion. We just investigated a microstructural combinatory including T-type micromixer, connecting pipe, and micropacked bed in sequence. All the elements can be obtained commercially with the key parameters presented in the manuscript already, such as the orifice diameter of T-type micromixer, I.D. and length of connecting tube, I.D, void fraction, empty volume of micropacked bed, morphology and size of packs. We provide some photos of the elements in Fig. 1 and make it more clear.
Besides, the novelty of our work comes from understanding and calibrating on the design of combinatory which is various and necessary for enhancing micromixing of viscous systems.
Correspondingly, the manuscript has been revised or illustrated.
Comment 2: Validation efforts are missing in this study. The authors should validate their setup and methodology. Also, uncertainties in the major parameters should be included.
Reply: Thank you for your suggestion. We agree that the validation of tool and methodology are fundamental, especially for electromechanical equipment and instrument. In our experiments, all the pumps and monitors were calibrated in advance to ensure to work well; all the measurements were repeated five times and averaged; and the error bars were shown in corresponding Figures.
Correspondingly, the manuscript has been revised or illustrated.
Comment 3: The authors should also include numerical results. In this format, flow physics is missing so that enhancement mechanisms could not be explained. Numerical results will give more clues for flow physics and performance enhancements.
Reply: Thank you for your suggestion. I agree that numerical results may give more clues of flow physics for performance enhancements. But, the micropacked-bed itself and the combinatory of commercial elements provide complicated spatial-structure and extended spatio-temporal scale for mixing flow, which rigorously challenge the numerical methodology in terms of geometric modelling and computational force. And, it is meaningless to sacrifice the accuracy of numerical simulation. In fact, it was lack of numerical simulation for micropacked bed in literature, and a few publications did not consider fine structure of micropacked bed and only discussed the heat transfer and pressure drop in homogeneous fluid without mixing. However, it is worth making efforts on developing numerical simulation methodology for complicated micro-structural system.
Correspondingly, the manuscript has been revised or illustrated.
Comment 4: The authors should include comparisons with the state of art micromixers. Some of passive ones including inertial ones are present in the literature. The authors could benefit from the technical content in such articles.
Reply: Thank you for your suggestion. I have added some introduction on the state of art of passive micromixers in the revised manuscript.
Comment 5: The trends in the experimental results (figs. 2-5) depend on the micromixer and are continuously changing. The changes in trends should be well explained.
Reply: Thank you for your suggestion. In general, the Xs decreased with the increasing of flow capacity continuously due to the enhancement of convection. As for other factors, continuously changing is difficult in practice, and we explained the difference under different conditions further in the revised manuscript.

Reviewer 2 Report
This work characterizes the micromixing in a system with T-type micromixer and micropacked bed in viscous reaction systems. According to the authors, the experimental validation from the micromixing laws enables guidance for the design and scaling-up of the combined T-type micromixer and micropacked bed towards micromixing intensification in viscous reaction systems.
In my opinion, this paper needs some revision. My comments are below.
C1: The introduction introduces a brief literature review on quantification of micromixing. In my opinion, some references are missing. Literature reports some works using reactive methods for micromixing quantification, for example:
Acid-base reaction
Fitschen, J.; Hofmann, S.; Wutz, J.; Kameke, A.v.; Hoffmann, M.; Wucherpfennig, T.; Schlüter, M. Novel evaluation method to determine the local mixing time distribution in stirred tank reactors. Chem. Eng. Sci. X 2021, 10, 100098.
Rodriguez, G.; Micheletti, M.; Ducci, A. Macro- and micro-scale mixing in a shaken bioreactor for fluids of high viscosity. Chem. Eng. Res. Des. 2018, 132, 890–901.
Eltayeb, A.; Tan, S.; Ala, A.A.; Zhang, Q. The study of the influence of slug density on the mixing performance in the reactor vessel, using PLIF experiment and FLUENT simulation. Prog. Nucl. Energy 2021, 131, 103558.
Ribeiro, J. P., Brito, M.S.C.A., Santos, R.J., Nunes, M.I. Reactive PLIF Method for Characterisation of Micromixing in Continuous High-Throughput Chemical Reactors. Processes. 2022, 10, 1916
Redox Reaction
Duan, X.; Feng, X.; Mao, Z.-S.; Yang, C. Numerical simulation of reactive mixing process in a stirred reactor with the DQMOM-IEM model. Chem. Eng. J. 2019, 360, 1177–1187.
Taghavi, M.; Moghaddas, J. Using PLIF/PIV techniques to investigate the reactive mixing in stirred tank reactors with Rushton and pitched blade turbines. Chem. Eng. Res. Des. 2019, 151, 190–206.
Hu, Y.; Wang, W.; Shao, T.; Yang, J.; Cheng, Y. Visualization of reactive and non-reactive mixing processes in a stirred tank using planar laser induced fluorescence (PLIF) technique. Chem. Eng. Res. Des. 2012, 90, 524–533.
C2: On page 1, line 42-44, it is mentioned “In particular, the low diffusion coefficient in high-viscosity resulted in slow molecular motion”. Authors should also comment on the change in flow regime with the viscosity changes.
C3: On page 2, line 47, there is a typo on the literature references.
C4: Equations 1-6, there is any coherence of the way how equations are introduced. Sometimes they are introduced as “:”, and other times they make part of the text (Equation 6).
C5: I would like that the authors comment on the impact of viscosity on the reaction rate constant (k1, k2 and k3). Equation 5 only depends on the temperature, and it does not depend on the viscosity. Was the empirical correlation defined for physical properties?
C4: Was Equation 9 demonstrated by the authors or proposed in the literature?
C5: On page 2, line 101, “platform” is not used as the right word. It should be “experimental setup”.
C6: On page 2, line 118, inside diameter and outside diameter should be defined as variables, e.g. di or do.
C7: On page 2, line 131, flow rate ratio R should be defined in italic, as a variable.
C8: On page 5, the authors showed that when the internal diameter is smaller, the micromixing performance is better. The authors should further explain the reason for this phenomenon.
C9: On page 5, I would expect to find some information on the flow regimes generated according to the fluids’ viscosity.
C10: A legend of Figure 6 is missing: the meaning of the black line, red dashed line, blue dashed line, and also the conditions of the experimental points.
Author Response
Responses to Reviewer’s comments
This work characterizes the micromixing in a system with T-type micromixer and micropacked bed in viscous reaction systems. According to the authors, the experimental validation from the micromixing laws enables guidance for the design and scaling-up of the combined T-type micromixer and micropacked bed towards micromixing intensification in viscous reaction systems.
In my opinion, this paper needs some revision. My comments are below.
Comment 1: The introduction introduces a brief literature review on quantification of micromixing. In my opinion, some references are missing. Literature reports some works using reactive methods for micromixing quantification, for example:
Acid-base reaction
Fitschen, J.; Hofmann, S.; Wutz, J.; Kameke, A.v.; Hoffmann, M.; Wucherpfennig, T.; Schlüter, M. Novel evaluation method to determine the local mixing time distribution in stirred tank reactors. Chem. Eng. Sci. X 2021, 10, 100098.
Rodriguez, G.; Micheletti, M.; Ducci, A. Macro- and micro-scale mixing in a shaken bioreactor for fluids of high viscosity. Chem. Eng. Res. Des. 2018, 132, 890–901.
Eltayeb, A.; Tan, S.; Ala, A.A.; Zhang, Q. The study of the influence of slug density on the mixing performance in the reactor vessel, using PLIF experiment and FLUENT simulation. Prog. Nucl. Energy 2021, 131, 103558.
Ribeiro, J. P., Brito, M.S.C.A., Santos, R.J., Nunes, M.I. Reactive PLIF Method for Characterisation of Micromixing in Continuous High-Throughput Chemical Reactors. Processes. 2022, 10, 1916
Redox Reaction
Duan, X.; Feng, X.; Mao, Z.-S.; Yang, C. Numerical simulation of reactive mixing process in a stirred reactor with the DQMOM-IEM model. Chem. Eng. J. 2019, 360, 1177–1187.
Taghavi, M.; Moghaddas, J. Using PLIF/PIV techniques to investigate the reactive mixing in stirred tank reactors with Rushton and pitched blade turbines. Chem. Eng. Res. Des. 2019, 151, 190–206.
Hu, Y.; Wang, W.; Shao, T.; Yang, J.; Cheng, Y. Visualization of reactive and non-reactive mixing processes in a stirred tank using planar laser induced fluorescence (PLIF) technique. Chem. Eng. Res. Des. 2012, 90, 524–533.
Reply: Thank you for your suggestion. We have added these literatures in the revised manuscript.
Comment 2: On page 1, line 42-44, it is mentioned “In particular, the low diffusion coefficient in high-viscosity resulted in slow molecular motion”. Authors should also comment on the change in flow regime with the viscosity changes.
Reply: Thank you for your suggestion. Increasing the viscosity individually will decrease the Reynold number, and inhibit the generation of convection. We have made revisions accordingly.
Comment 3: On page 2, line 47, there is a typo on the literature references.
Reply: Thank you for your notification. We have made revision.
Comment 4: Equations 1-6, there is any coherence of the way how equations are introduced. Sometimes they are introduced as “:”, and other times they make part of the text (Equation 6).
Reply: Thank you for your notification. We have made revision. The equations are introduced as “:”.
Comment 5: I would like that the authors comment on the impact of viscosity on the reaction rate constant (k1, k2 and k3). Equation 5 only depends on the temperature, and it does not depend on the viscosity. Was the empirical correlation defined for physical properties?
Reply: Thank you for your comment. The empirical correlation did not contain physical properties. It is truly an important issue. We supposed the viscosity of inert media did not affect the intrinsic reaction kinetics. This was also proposed in some literatures.
Comment 6 : Was Equation 9 demonstrated by the authors or proposed in the literature?
Reply: Equation 9 can be derived from aforementioned equations, and it was proposed in the literature as well.
Comment 7: On page 2, line 101, “platform” is not used as the right word. It should be “experimental setup”.
Reply: The revision has been made.
Comment 8: On page 2, line 118, inside diameter and outside diameter should be defined as variables, e.g. di or do.
Reply: Thank you for your suggestion. Considering the variables in this work are numerous and too many definitions may cause chaotic, we do not define inside diameter and outside diameter specifically.
Comment 9: On page 2, line 131, flow rate ratio R should be defined in italic, as a variable.
Reply: The revision has been made.
Comment 10: On page 5, the authors showed that when the internal diameter is smaller, the micromixing performance is better. The authors should further explain the reason for this phenomenon.
Reply: Thank you for your suggestion. As for a microchannel for mixing, the decease of internal diameter will decrease the mass transfer distance and accelerate the mixing. We have added explanation in the revised manuscript.
Comment 11: On page 5, I would expect to find some information on the flow regimes generated according to the fluids’ viscosity.
Reply: According to the fluids’ viscosity and the flow capacity, the Reynold number was far less than 2000. Judged from the common criterion, the flow pattern was always laminar flow. Certainly, it is still difficult to determine the flow pattern experimentally.
Comment 12: A legend of Figure 6 is missing: the meaning of the black line, red dashed line, blue dashed line, and also the conditions of the experimental points.
Reply: Thank you for your suggestion. We have added illustrations in the caption.

Round 2
Reviewer 1 Report
The authors addressed most of the comments of the reviewer. The reviewer is generally content with the revision.
Reviewer 2 Report
I recommend the publication of the work "Micromixing intensification within a combination of T-type micromixer and micropacked bed" in Micromachines.